# Shape morphing of hydrogels by harnessing enzyme enabled mechanoresponse

Kuan Zhang [1,2,3,5], Yu Zhou[2,5], Junsheng Zhang[1], Qing Liu[1], Christina Hanenberg[2,3], Ahmed Mourran [2], Xin Wang[1], Xiang Gao[2,3], Yi Cao [1,4], Andreas Herrmann [2,3] & Lifei Zheng [1]

Hydrogels have been designed to react to many different stimuli which find broad applications in tissue engineering and soft robotics. However, polymer networks bearing mechano-responsiveness, especially those displaying on-demand self-stiffening and self-softening behavior, are rarely reported. Here, we design a mechano-controlled biocatalytic system at the molecular level that is incorporated into hydrogels to regulate their mechanical properties at the material scale. The biocatalytic system consists of the protease thrombin and its inhibitor, hirudin, which are genetically engineered and covalently coupled to the hydrogel networks. The catalytic activity of thrombin is reversibly switched on by stretching of the hydrogels, which disrupts the noncovalent inhibitory interaction between both entities. Under cyclic tensile-loading, hydrogels exhibit self-stiffening or self-softening properties when substrates are present that can self-assemble to form new networks after being activated by thrombin or when cleavable peptide crosslinkers are constitutional components of the original network, respectively. Additionally, we demonstrate the programming of bilayer hydrogels to exhibit tailored shape-morphing behavior under mechanical stimulation. Our developed system provides proof of concept for mechanically controlled reversible biocatalytic processes, showcasing their potential for regulating hydrogels and proposing a biomacromolecular strategy for mechano-regulated soft functional materials.

The regulation of enzyme activity through different signaling pathways not only controls cellular metabolic processes[1,2] but also plays important roles during tissue regeneration and destruction[3,4]. Inspired by these natural systems, chemists have striven to develop artificial enzyme-based stimuli-responsive systems for multiple applications[5,6]. Among these, temperature[7], pH[8], small molecules[9], and light[10,11] were widely applied as stimuli for controlling enzymatic activities. In contrast, mechanical force was rarely used in this context, resulting from the challenges of converting mechanical force into biochemical

signals[12–14]. Our group recently demonstrated the application of ultrasound, which indirectly induces a force on biomacromolecular chains and assemblies, to reversibly and irreversibly control biocatalytic reactions in solution[12,13]. Mertz et al. reported reversible on/off switching of enzyme activity by mechanically stretching multi-layered polyelectrolyte films where enzymes were incorporated[14]. Although this mimetic of cryptic site-bearing proteins showed a similar cause-consequence effect, the process underlying this design was largely different in terms of scale from that occurring in proteins (i.e., material

[1]Wenzhou Institute, University of Chinese Academy of Sciences, Wenzhou 325001, China. [2]DWI – Leibniz-Institute for Interactive Materials, Aachen 52056, Germany. [3]Institute for Technical and Macromolecular Chemistry, Rheinisch-Westfälische Technische Hochschule (RWTH) Aachen University, Aachen 52074, Germany. [4]Collaborative Innovation Center of Advanced Microstructures, National Laboratory of Solid State Microstructure, Department of Physics, Nanjing University, 22 Hankou Road, Nanjing 210093, China. [5]These authors contributed equally: Kuan Zhang, Yu Zhou. ✉e-mail: herrmann@dwi.rwth-aachen.de; zhenglf@ucas.ac.cn

scale *vs* molecular scale). Moreover, the demonstration of the capability of mechanically triggered biochemical reactions that can further regulate the physical-chemical properties of bulk materials is lacking.

Fascinated by enzymes involved in mechano-transduction that transform mechanical forces into chemical processes and with the goal of creating mechanically responsive tissue-like materials[15], we designed biologically inspired hydrogels, which represent one of the most promising material candidates to tissue-tissue or tissue-electronic interfaces, in light of their biological, mechanical, chemical, and physical similarities to different tissues[16–18]. We investigated whether the control over enzyme activity by mechanical force at the molecular scale can be tailored to regulate the mechanical properties of hydrogels at the material scale. In this context, recent research

efforts focus on the development of self-stiffening hydrogels under mechanical stress using mechanophores[19–21], nanoparticles[22], and crystalline structures[23,24]. However, due to the radicals generated through mechanical actuation and the use of toxic molecules, the reported systems still suffer from slow regulation, poor controllability, and biocompatibility[19,21,25,26]. Moreover, all these efforts aimed at unilaterally enhancing the mechanical properties of the networks. Systems that can achieve both self-stiffening or self-softening on demand were scarcely reported[27].

In this work, we introduce a controllable biocatalytic activation system through cyclic mechanical loading for rapid hydrogel regulation (Fig. 1a). Thrombin, a protease that is involved in blood coagulation and catalyzes the formation of fibrin by digestion of a defined peptide sequence, and its inhibitor, hirudin, were selected. The

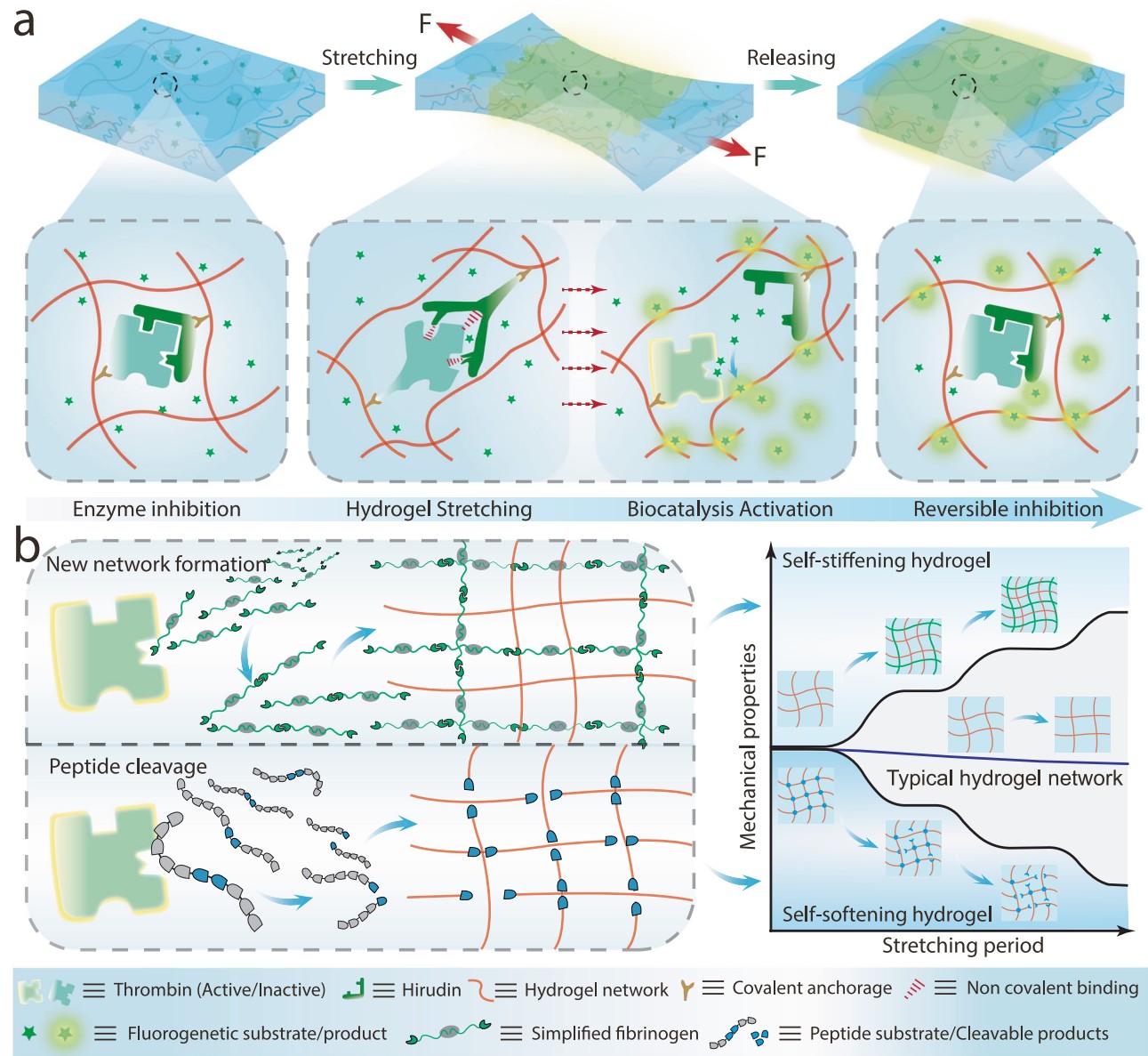

**Fig. 1 | Mechano-controlled reversible biocatalysis for hydrogel regulation. a** Schematic illustration of the controllable biocatalytic system in the hydrogel matrix. The enzyme and its inhibitor were incorporated into the hydrogel network by covalent bonds, and the catalytic activity was activated by mechanical stretching of the hydrogel. **b** Schematic illustration of the mechanism of the regulation of the

mechanical properties of hydrogel through controlled biocatalysis. A substrate that can be activated by thrombin to self-assemble into a second network is added to the hydrogel, resulting in the self-stiffening behavior of the polymer networks. Alternatively, a self-softening hydrogel can be obtained when a thrombin-cleavable crosslinker is introduced.

genetically engineered thrombin–hirudin enzyme pair was covalently incorporated into hydrogel networks through the introduction of double bonds at defined positions. The biocatalytic activity can be switched on and off reversibly by breaking or healing the non-covalent interaction between thrombin and hirudin upon applying or removing the mechanical stretching on the hydrogel. The presence of additional peptide substrates encoding specific thrombin cleavage sites enabled mechano-regulated hydrogels with self-stiffening or self-softening behaviors under repetitive stretch-release cycles (Fig. 1b). Furthermore, these two types of hydrogels were exploited in synergy to prepare bilayer hydrogel actuators, which exhibited complex shape-morphing behavior from programmed 2D structures to 3D objects upon mechanical stimulation. We anticipate that our approach will find a wide range of applications in areas such as drug delivery, tissue engineering, and soft robotics[28].

## Results

### Design and function of the mechano-sensitive system

Key to our design is a specific enzyme/inhibitor interaction, which can be separated by force and hence is responsive to mechanical stimulation. For this purpose, thrombin and its inhibitor, hirudin, were chosen based on the following considerations. First, hirudin can form a stable complex with thrombin through multiple noncovalent interactions, resulting in the deactivation of thrombin. Second, inspired by the recent studies on similarly inhibited enzymes that could be activated by ultrasound in solution[12], we hypothesized that mechanical stretching of a hydrogel with covalently incorporated protein units might also be suitable to break the interaction between the embedded enzyme and inhibitor and consequently activate the catalytic activity.

In order to integrate thrombin and hirudin into the hydrogel network covalently, we used the well-established S-nitroso-N-acetylpenicillamine (SNAP) chemistry[29,30]. SNAP protein was fused to the C-terminus of both prethrombin-2 and hirudin during plasmid constructions (Supplementary Fig. 1a). Unfortunately, prothrombin-2-SNAP was expressed as inclusion bodies. Therefore, a denaturation and refolding step was performed to obtain the soluble folded protein. Subsequently, the refolded prethrombin-2-SNAP fusion was digested with Factor Xa to obtain the desired thrombin-SNAP (Thr-S, Supplementary Fig. 1b). The expression of soluble hirudin-SNAP (Hir-S) was achieved by fusing a small ubiquitin-related modifier (SUMO) protein to the N-terminus of hirudin, which could be readily removed by SUMO protease during the purification step (Supplementary Fig. 1c). Sodium dodecyl sulfate-polyacrylamide gel electrophoresis (SDS-PAGE) analysis and matrix-assisted laser desorption/ionization time-of-flight mass spectrometry (MALDI-TOF-MS) results confirmed the purity and identity of Thr-S and Hir-S (Supplementary Figs. 2–4). Then, O6-benzylguanine styrene (BS) was synthesized and reacted with the SNAP-tagged proteins to produce thrombin and hirudin with a vinyl group (Thr-V and Hir-V, Fig. 2a and Supplementary Figs. 6–9).

Next, we evaluated the binding affinity between Thr-S and Hir-S with isothermal titration calorimetry (ITC). As can be seen in Fig. 2f, the dissociation constant ($K_d$) was ~$10^{-9}$ M, which is much lower than most protein interactions ($10^{-4} < K_d < 10^{-8}$ M). To further measure the enzyme–inhibitor interaction, we used atomic force microscopy (AFM)-based single-molecule force spectroscopy (SMFS) to explore the dissociation mechanism between Thr-S and Hir-S (Fig. 2b). The variation of rupture forces under different pulling rates of applied forces was measured to obtain kinetic force spectra. As shown in Fig. 2c–e and Supplementary Fig. 10, the rupture force of the interacting Thr-S and Hir-S was centered at ~50 pN at a pulling speed of 200 nm s$^{-1}$ and increased gradually with the pulling rate, indicating that the rupture events occurred under non-equilibrium conditions according to the Bell–Evans model[31,32]. Meanwhile, the rupture distance $\Delta x_u$ of 0.29 nm and the spontaneous off rate at zero force ($\alpha_0$) of 2.60 s$^{-1}$ were calculated using the two-state Bell–Evans model fitting

parameters. Notably, the obtained rupture distance is roughly consistent with the length of the hydrophobic interactions shown in reported structures[32]. Based on these results, it is reasonable to expect that the interaction between the embedded enzyme and inhibitor in hydrogel could be regulated by mechanical forces applied at the macroscopic level of the polymer network.

Next, the catalytic activities of Thr-S with or without Hir-S were investigated by using a fluorogenic substrate (Supplementary Fig. 11a). As shown in Fig. 2g, in the absence of the inhibitor, Thr-S catalyzed the cleavage of the substrate, producing a 5(6)-carboxyfluorescein-bearing peptide fragment with strong fluorescence at 450 nm. When Hir-S was introduced, the fluorescent signal was significantly weakened. Detailed study on the catalytic reaction at varying Hir-S-to-Thr-S ratios revealed that a complete inhibition on the activity of thrombin was achieved when the ratio was higher than two (Supplementary Fig. 11b). Therefore, these results demonstrated that both proteins remained functional after the recombinant modification.

### Mechano-controlled reversible biocatalysis

Considering the broad solvent compatibility and oxygen tolerance, we utilized well-established thiol-ene chemistry to prepare the hydrogels[33]. The cross-linking of 4-arm poly(ethylene glycol) (PEG) thiol, PEG diacrylate (PEGDA) and the preincubated Thr-V/Hir-V or Thr-S/Hir-S complexes ($n_{Thr-S}$:$n_{Hir-S}$ = 1:2.5) was initiated by lithium phenyl-2,4,6-trimethylbenzoylphosphinate (LAP) under UV light irradiation for 20 min (Supplementary Fig. 13a). The hydrogels had an average thickness of about 1 mm. The mechanical properties of the hydrogels were assessed using a tensile test with a constant stretch rate of 10 mm/min at room temperature in air using a universal testing machine. The typical stress–strain curves of the hydrogels are shown in Supplementary Fig. 13b. The hydrogel was highly stretchable, showing a maximum extension ratio of more than four times its original length.

Firstly, to investigate the impact of restricted diffusion within hydrogel networks on enzyme activity, we measured the reaction rates of thrombin in both phosphate-buffered saline (PBS) solution and hydrogels. As shown in Supplementary Fig. 14 (red and black lines), the thrombin activity within the hydrogels was found to be approximately 85% compared to the activity observed in the PBS solution. These results indicate that the restricted diffusion within the hydrogel network did not significantly affect the enzyme activity. One possible explanation for this observation is the nature of the fluorogenic substrate used in this study, which had a relatively small size (MW = 624 g/mol). However, when using larger enzyme substrates, we can anticipate a decrease in enzyme activity due to the limited diffusion within the hydrogel network.

To demonstrate the activation of thrombin through hydrogel stretching, the fluorogenic peptide substrate was added during the preparation of hydrogels. A picture of the hydrogel containing the Thr-V/Hir-V complex before and after stretching with 150% strain is shown in Fig. 3a. After stretching, a brighter area appeared in the center region of the specimen. Furthermore, a portable fluorimeter was used to measure the emission spectra before and after tensile tests. A fluorescence signal was detected after stretching the same hydrogel, preliminarily confirming the mechanical activation of thrombin and efficient hydrolysis of fluorogenic molecules in this matrix after mechanical actuation. In the control gel containing the Thr-S/Hir-S complex, no variation in fluorescence emission was detected before and after stretching, demonstrating the importance of the covalent linkage of proteins to the hydrogel network for efficient mechanotransduction during stretching (Fig. 3b).

Next, the relationship between the efficiency of thrombin activation and the stretching strain was investigated by recording the fluorescence emission spectra of the hydrogels after stretching to certain degrees for 3 min (Fig. 3b). The fluorescence intensities at 450 nm measured at different positions of each sample were averaged

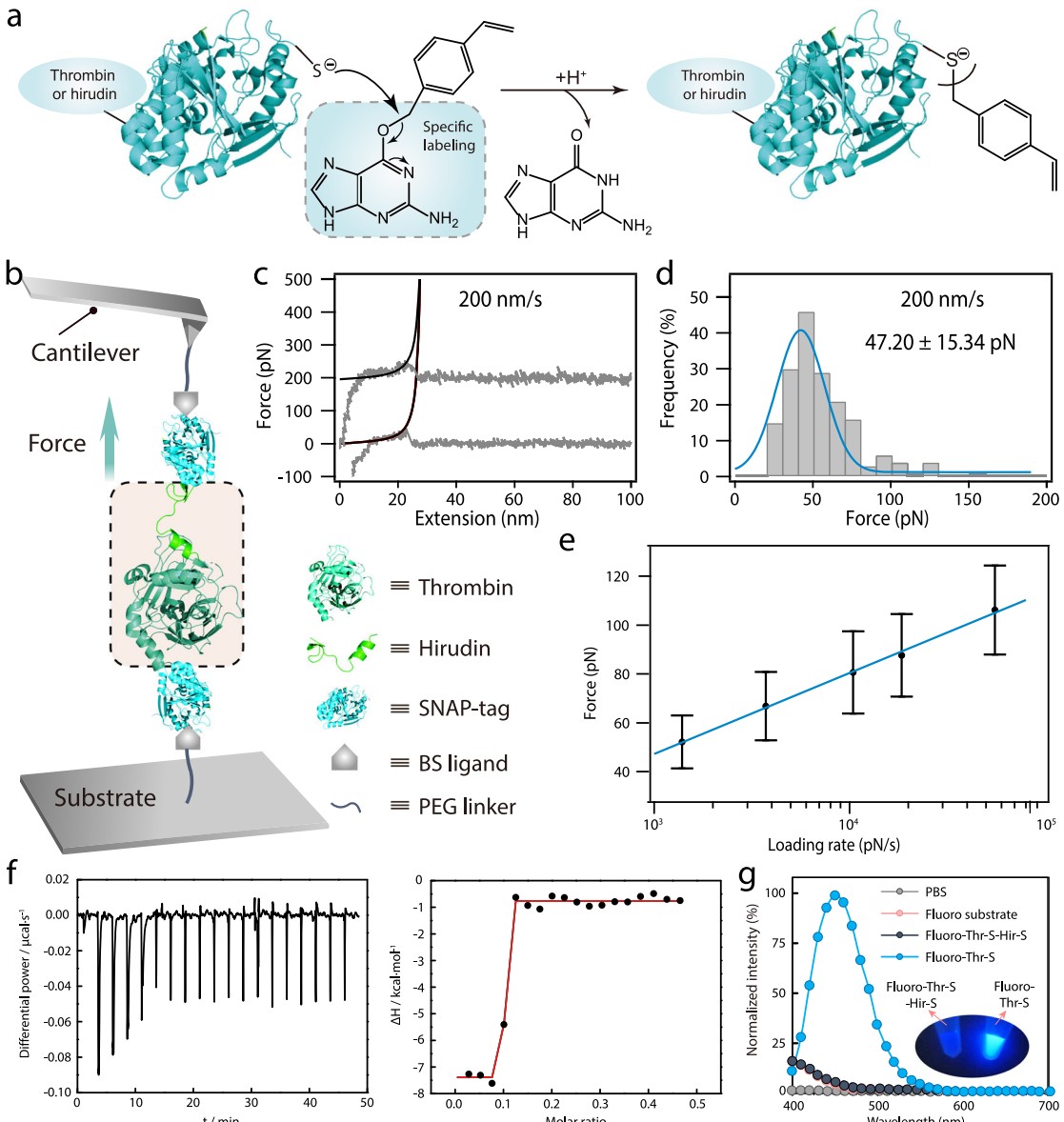

**Fig. 2 | Synthesis and characterization of the enzyme-inhibitor system. a** The reaction scheme of introducing a vinyl group to Thr-S and Hir-S with BS ligand. **b** Schematic illustration of AFM-based single-molecule force measurements. Thr-S and Hir-S were linked to the BS-modified probe and substrate, respectively. **c** Typical force–retract curve for the rupture of the Thr-S and Hir-S at a pulling speed of 200 nm/s (gray lines). Worm-like chain fitting of the force–extension curves (black lines) confirmed that the peak at an extension range of 10–30 nm corresponds to the rupture of the Thr-S/Hir-S interaction. **d** Rupture force histograms at a pulling speed of 200 nm/s. The Gaussian fitting shows an average rupture force of 47.20 ± 15.34 pN. **e** The rupture forces of the interactions between Thr-S and Hir-S at different pulling speeds. The rupture forces increased gradually as the pulling rates became larger. **f** ITC characterization of the Thr-S/Hir-S binding. The tests were conducted in PBS buffer (pH = 7.4), the Thr-S concentration in the bath was 26.2 μM, and the Hir-S concentration in the syringe was 108 μM. **g** Tests of the catalytic activities of Thr-S in the presence or absence of Hir-S using a fluorogenic substrate. In the absence of an inhibitor, Thr-S catalyzed the cleavage of the substrate, producing strong fluorescence, as shown in the inset picture. Data are presented as mean values with error bars representing the standard deviation of three independent replicates (*n* = 3 in each group). Source data are provided as a Source Data file. Fluoro, Fluorogenic. Thr-S, thrombin-SNAP. Hir-S, Hirudin-SNAP.

and used for the following comparison. As shown in Fig. 3c, no significant changes in fluorescence were observed below a critical stretching strain of 100%. However, the fluorescence increased significantly when the stretching strain reached 120%. At more than 150% strain, a plateau was reached. To estimate the dissociation ratio of the protein complex in the stretched gel, we measured the reaction rates of stretched gel with free thrombin and hirudin-inhibited thrombin. By comparing the reaction rates shown in Supplementary Fig. 15, the dissociation ratio of the thrombin–hirudin complex at a stretching strain of 150% was estimated to be around 19%. To test if the activity of thrombin can be regulated reversibly, the increase of fluorescence was

measured by alternating stretched (150% strain) and unstretched states of the hydrogel multiple times. As shown in Fig. 3d, the fluorescence remained unchanged before stretching the hydrogel, but after each stretching cycle, the fluorescent signal increased. Once the hydrogel returned to the unstretched state, the fluorescent signal remained at the same level. These results indicated that the hydrogel became enzymatically active as soon as the critical stretching degree was reached and that the activity was inhibited as soon as the hydrogel returned to the unstretched state.

Finally, to investigate the stability of the materials, specifically the stability of enzymes in the hydrogel, we measured the reaction rates of

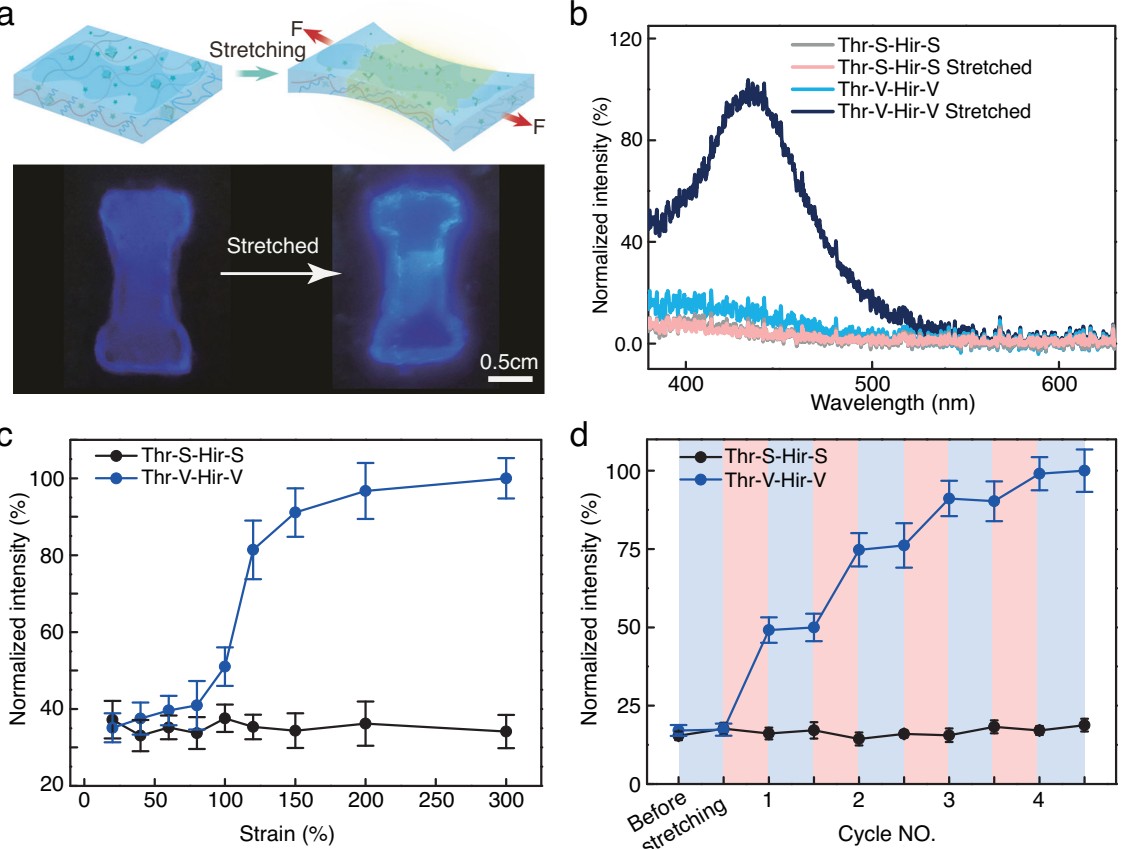

**Fig. 3 | Mechano-controlled reversible biocatalysis. a** Schematic illustration of fluorescence generation based on the activation of thrombin through mechanical stretching (top). Picture of the hydrogel before and after stretching (bottom). The fluorescence mainly appears in the stretching region. **b** Emission spectra of the fluorogenic substrate inside hydrogels before and after mechanical stretching. **c** Fluorescence intensities measured in hydrogels for various stretching strains ranging from 20% to 300%. **d** Measurements of the fluorescence of alternatingly stretched (150% strain) and unstretched hydrogels (the time intervals for all tests are 3 min). Each experiment was repeated three times independently with similar results. Data are presented as mean values with error bars representing the standard deviation of three independent replicates ($n = 3$ in each group). Source data are provided as a Source Data file. Thr-S thrombin-SNAP, Hir-S Hirudin-SNAP, Thr-V thrombin-SNAP-Vinyl, Hir-V hirudin-SNAP-Vinyl.

thrombin using the fluorogenic substrate after storing the hydrogels for 0, 3, 7, and 14 days at 4 °C. As shown in Supplementary Fig. 14, the activity of thrombin decreased over time. After a period of 14 days, the activity of thrombin in the hydrogels had declined to below 50% compared to the initial activity of the freshly prepared hydrogels. These results indicate the occurrence of certain degrees of denaturation and/or degradation of embedded enzymes in hydrogels. Further research regarding the use of enzyme stabilizers or additives that protect the enzyme's structure and activity can help prolong its functional lifespan.

## Mechanically-triggered self-stiffening or self-softening hydrogels

After activation of biocatalysis inside hydrogels through stretching, we wished to exploit the mechano-responsive catalytic system to regulate the mechanical properties of hydrogels. To realize a self-stiffening hydrogel, fibrinogen was chosen as an embedded substrate during the preparation of the hydrogel since it can be activated by thrombin to form fibrin fibers (Supplementary Figs. 17–21). As shown in Supplementary Fig. 18, the increase in absorbance at 650 nm confirmed the successful conversion of fibrinogen into fibrin fibers catalyzed by Thr-S in solution. Subsequently, we investigated whether the generation of the fibrous network can strengthen the material in a way reminiscent of natural muscles. For this purpose, cyclic tensile tests were carried out and each cycle consisted of three steps: first, the hydrogel in the

initial state was stretched at a speed of 10 mm/min; then, the strain was maintained at 150% for 180 s; finally, the hydrogel was brought back to the unstretched state. As shown in Fig. 4a, b, the mechanical strength of the hydrogels increased by ~70% after five tensile cycles. In contrast, no obvious change in the mechanical property was observed for the control hydrogels containing the Thr-S/Hir-S complex, indicating that the activation of thrombin was essential for the self-stiffening behavior of the hydrogel. To examine the effect of hydrogel storage time on fibrinogen stability, we measured the self-stiffening property of the hydrogels after storage for 0–14 days. The results showed that the improvement of the mechanical properties of the hydrogels after training sharply decreased after day 7 (Supplementary Fig. 22). This suggests that the fibrinogen remains relatively stable within the hydrogels for up to 7 days.

Next, in order to endow hydrogels with self-softening behavior, poly(sodium acrylate) hydrogels containing thrombin-cleavable peptide (LVPRGS) as an additional crosslinker were prepared (Supplementary Fig. 23). It is worth noting that while PEG-based hydrogel systems are also suitable for self-softening research, we considered the cost implications of replacing PEGDA with the thrombin-cleavable peptide crosslinker. The resulting poly(sodium acrylate) hydrogels exhibited a maximum strain close to 600% (Supplementary Fig. 26). To confirm the activation of thrombin activity within these hydrogels, we conducted similar experiments using the same fluorogenic substrate as mentioned above. The results revealed that

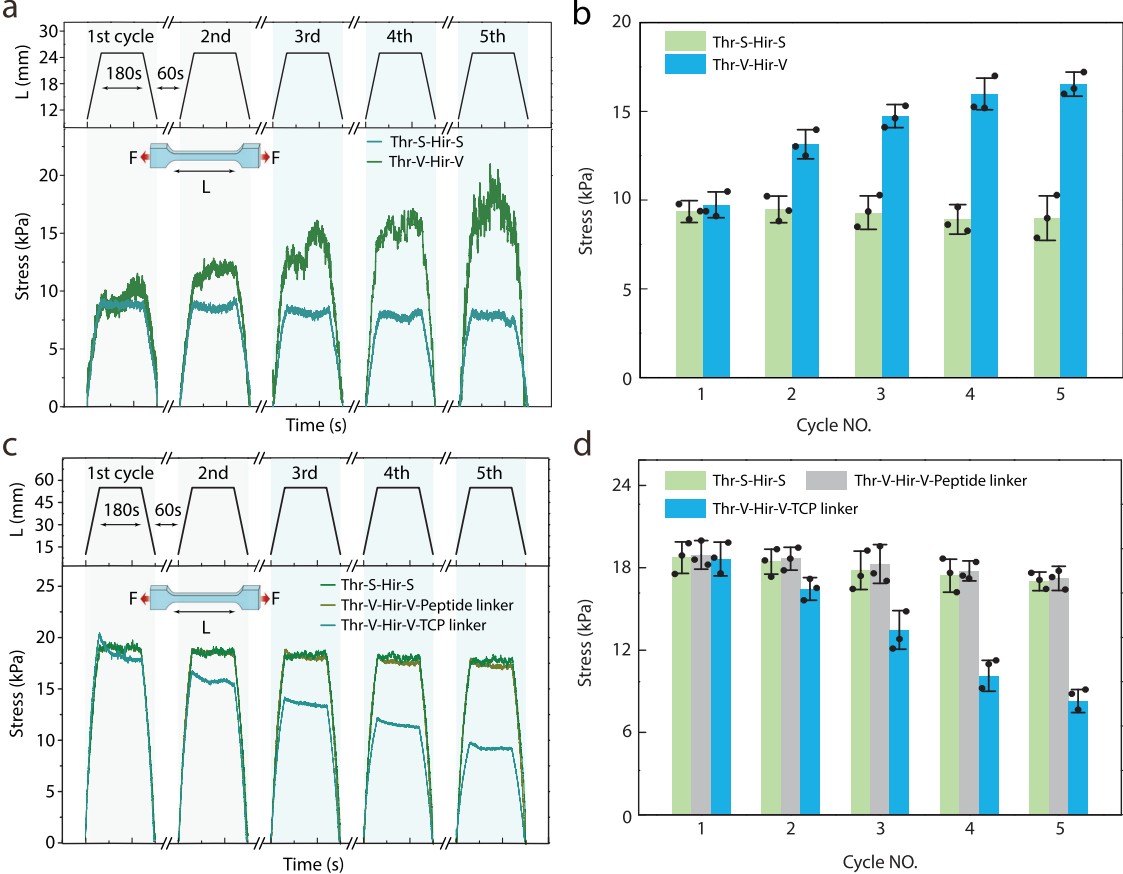

**Fig. 4 | Mechanically triggered self-stiffening or self-softening of hydrogels.**
**a** Self-stiffening behavior of the hydrogels under repetitive mechanical stretching. The thrombin and hirudin used in control groups are not modified with a double bond and, hence, are not connected to the polymer network. **b** Summary of the mechanical properties of the self-stiffening and control hydrogels after each tensile cycle. **c** Self-softening behavior of the hydrogels under repetitive mechanical stretching. In control group 1, the thrombin and hirudin used are not modified with a double bond. In control group 2, the crosslinker was a thrombin non-cleavable

peptide. **d** Summary of the mechanical properties of the self-softening and control hydrogels after each tensile cycle. Each experiment was repeated three times independently with similar results. Data are presented as mean values with error bars representing the standard deviation of three independent replicates ($n = 3$ in each group). Source data are provided as a Source Data file. Thr-S thrombin-SNAP, Hir-S Hirudin-SNAP, Thr-V thrombin-SNAP-Vinyl, Hir-V hirudin-SNAP-Vinyl, TCP thrombin cleavable peptide.

the fluorescence intensity increased when the stretching strain exceeded 100% and that the fluorescence signal increased proportionally to the strain of the hydrogel (Supplementary Fig. 27). Subsequently, we investigated the possibility of regulating thrombin activity in a controlled manner within poly(sodium acrylate) hydrogels. As illustrated in Supplementary Fig. 28, the fluorescence signals increased when the hydrogels were stretched to 150% for 180 s and maintained their signal strength after being released. This result was consistent with the mechano-triggered self-stiffening hydrogel we developed and thus demonstrated the general applicability of our principle for the design of hydrogels consisting of different constituent building blocks to activate the target enzyme and control its activity in response to mechanical force. To further evaluate the influence of the catalytic process on the mechanical properties of hydrogels, cyclic tensile tests were performed. As control groups, hydrogels containing thrombin non-cleavable peptide crosslinkers or Thr-S/Hir-S complexes were fabricated. Initially, we tested with a strain of 150%, but only a slight difference between the control and mechano-activatable group was observed (Supplementary Fig. 29a). We then increased the strain to 450% and noticed that the latter group appeared to loosen compared to the control group (Supplementary Fig. 29b) indicating that the decreased mechanical properties of the hydrogels were mainly due to the thrombin-cleavable peptide crosslinker being cut upon stretching

the hydrogels. Further increasing the Thr-V/Hir-V complex concentration resulted in ~50% decrease in the mechanical properties of the hydrogels after five tensile cycles compared to the pristine hydrogels (Fig. 4c, d). Taken together, these findings demonstrated the successful transduction of the reversible control of enzyme activity at the molecular level to the regulation of mechanical properties of macroscopic hydrogels at the material level.

**Programming shape morphing behavior of bilayer hydrogels**
Bilayer shape-morphing hydrogels offer great potential for a range of applications owing to the asymmetric responsive behaviors of their two hydrogel layers, which possess different swelling or mechanical properties[34]. Our developed enzyme-regulated hydrogels represent a promising avenue for designing shape-morphing hydrogels, as their mechanical properties can be controlled by an external mechanical stimulus. The combination of self-stiffening and self-softening hydrogels can be exploited to program the shape-morphing properties of bilayer hydrogels in a controlled and predictable manner.

To validate our principles for engineering shape-morphing hydrogels based on mechanical protein activation, we utilized the mechano-responsive polymer networks to prepare bilayer hydrogels with one layer exhibiting self-stiffening and the other layer self-softening properties using a custom-made mold (Fig. 5a). In order to

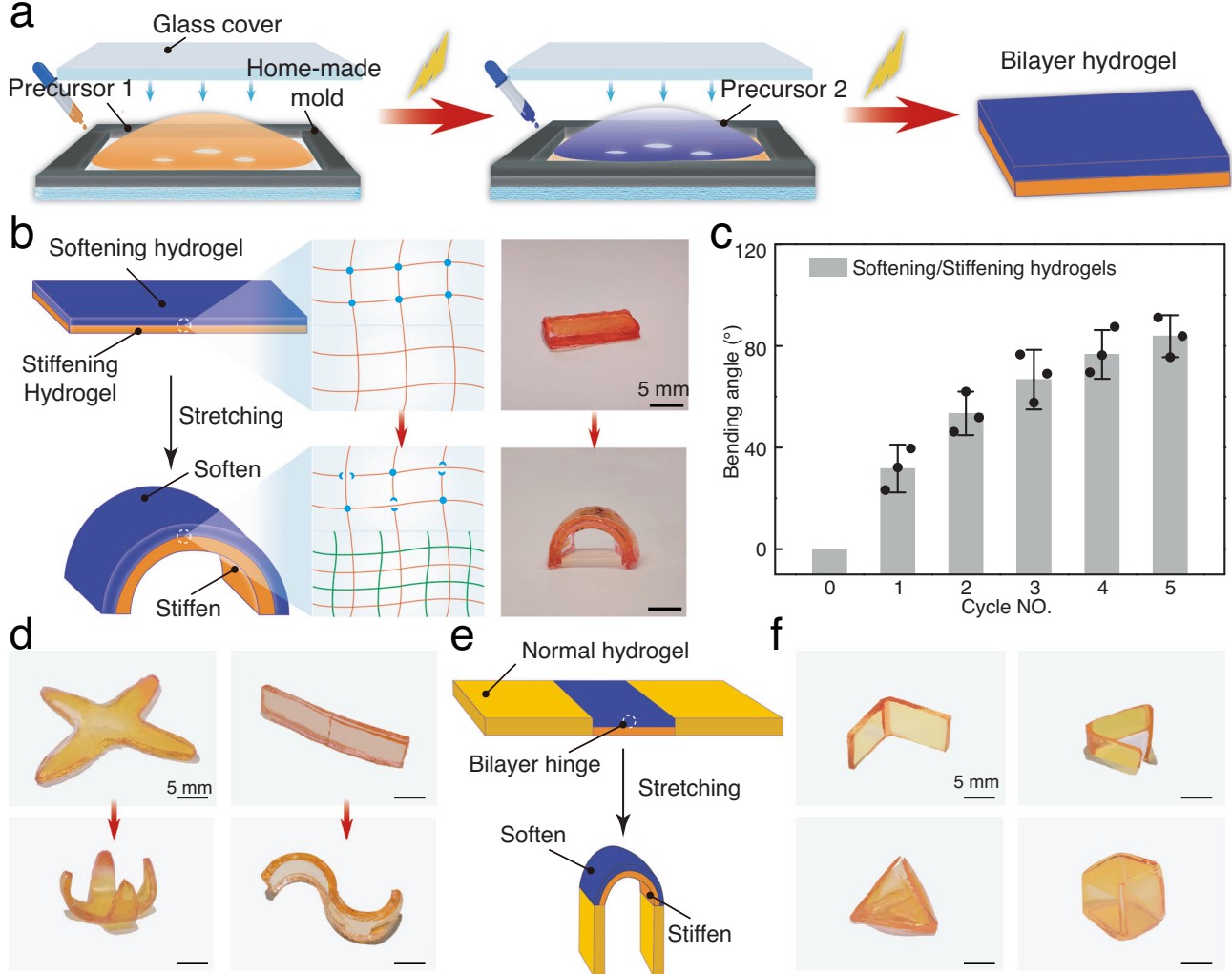

**Fig. 5 | Programming shape morphing behavior of bilayer hydrogels.**
**a** Schematic illustration of the preparation of the bilayer hydrogels through the layer-by-layer process using a custom-made mold. **b** Schematic presentation of the mechanism of shape morphing behavior of the bilayer hydrogels based on mechanically controlled self-stiffening/self-softening properties of constituent hydrogels (left). A picture of the bilayer hydrogel consisting of self-stiffening/self-softening hydrogel layers before and after mechanical stretching (right). **d** Photographs of the programmed 2D hydrogel strips, which transformed into distinct 3D architectures, including a 4-petal flower shape and S-shape upon

stretching. **c** The evolution of bending angles of the bilayer hydrogels from 0 to 85° through different numbers of tensile cycles. **e** Schematic illustration of the transformation of the bilayer hydrogels containing hinges under mechanical stretching. **f** Photographs of programmed 2D hydrogel structures, which transformed into distinct 3D architectures. Each experiment was repeated three times independently with similar results. Data are presented as mean values with error bars representing the standard deviation of three independent replicates ($n = 3$ in each group). Source data are provided as a Source Data file.

prevent the bilayer hydrogels from bending prior to stretching, both layers need to possess similar mechanical properties. Therefore, we prepared a series of PEG and poly(sodium acrylate) hydrogels with different mechanical properties by adjusting the monomer content. Based on the results shown in Supplementary Fig. 30, we chose PEG hydrogels with 6 *wt.%* and poly(sodium acrylate) hydrogels with 40 wt %. The two layers were covalently bonded together due to the presence of unreacted monomers during the preparation of the first layer. The bilayer hydrogel retained its straight strip-like shape upon removal from the mold, and a clear boundary between the two layers was visible. Moreover, no delamination between the two layers was observed in tensile testing, enabling shape morphing (Fig. 5b). Then, bilayer hydrogels were stretched with an extension ratio of 300% for shape morphing experiments. After the removal of strain, the bilayer hydrogel spontaneously bent towards the side of the mechanically enhanced layer, achieving a bending angle of 95 ± 14° (Supplementary Fig. 32, Fig. 5b, d). We could regulate the bending angles of the bilayer hydrogels from 0 to 85° through different tensile cycles by controlling

the activation of thrombin (Fig. 5d). Using the self-stiffening or softening hydrogel as one layer and mechanically non-activatable hydrogel without enzyme as the other layer, we could regulate the bending angles of the bilayer hydrogels from 0 to 50° (Supplementary Fig. 33). This result clearly indicates that both mechano-sensitive hydrogel layers are important to obtain maximum shape morphing properties and that they act in synergy. To investigate the effect of layer thickness on bending behaviors, we prepared bilayer hydrogels with various layer thicknesses. As shown in Supplementary Fig. 34, the bending angle of the bilayer hydrogel decreased from -115° to -30° with an increase in the total thickness. Our results demonstrate that we could rationally tune the bending angles of bilayer hydrogel strips by varying the thickness, the mechanical properties of the constituent hydrogel components, and the mechanical actuation of the two-hydrogel layer system itself.

The tunable bending behavior of the hydrogel, along with the significant bending angle achieved, qualifies these materials to achieve more complex, controllable, and programmable shape morphing

under mechanical triggering. Therefore, we first programmed 2D hydrogel strips into distinct 3D architectures, including a 4-petal flower shape, a bowknot shape, and an S-shape, which were fabricated to exhibit a bilayer hydrogel architecture. Stretching the bilayer hydrogels in different directions led to the formation of spiral shapes (Fig. 5d and Supplementary Fig. 35). More complex shape-morphing can be realized by combining mechano-activatable and static hydrogel components in a single specimen (Fig. 5e). The bilayer units can be used as local hinges that drive the transformation of the entire hydrogel in response to mechanical stimuli. To prove this, we fabricated a strip hydrogel containing one or two anisotropic enhancing/losing bilayer units and conventional hydrogel units without enzyme. The bilayer hinge bent after stretching, while the parts lacking the enzyme remained linear, leading to an overall deformation (Fig. 5f). The location of bending could be controlled by deliberately patterning the two units. Furthermore, the bending angle could be precisely programmed by adjusting stretching time and hydrogel thickness, enabling the programming of more sophisticated shape-morphing behaviors. Based on this, we were able to build several 3D shape hydrogel objects, indicating that the force generated by the bilayer deformation could be exploited to achieve complex 3D shape morphing starting from 2D hydrogel structures (Fig. 5f and Supplementary Fig. 36).

## Discussion

In this study, we present a mechano-controlled biocatalytic system for rapid hydrogel regulation. Genetically engineered thrombin and hirudin were incorporated into the hydrogel matrix, and on/off switching of thrombin activity was achieved by breaking or healing the non-covalent interaction between thrombin and its inhibitor hirudin using mechanical force. Furthermore, the reversible control of enzyme activity at the molecular level was exploited to regulate the mechanical properties of the hydrogels at the material level. Under cyclic tensile loading, the developed hydrogels exhibited self-stiffening or self-softening behaviors. This represents a biomacromolecular strategy for force-induced reinforcement or attenuation of hydrogel networks. In addition, we fabricated a bilayer hydrogel consisting of the two aforementioned hydrogels and achieved shape-morphing behavior after mechanical stimulation. Through precisely programming the shape-morphing behavior, we were able to create various 3D hydrogel objects by incorporating the stiffening/softening bilayer units as hinges to drive the overall hydrogel transformation. Furthermore, the complexity of shape-morphing can be expanded by combining mechano-active hydrogel units with conventional ones. We anticipate that these results will pave the way for the development of self-regulating materials, which have potential applications in artificial muscles, soft robotics, and in vivo implants requiring tunable mechanical properties.

## Methods

### Plasmids construction

The fragments containing prethrombin 2-SNAP and SUMO-Hirudin-SNAP were ligated to pJET 1.2/blunt vector using T4 DNA ligase in a 3:1 molar ratio according to the blunt-end ligation protocol. After 30 min of incubation at 16 °C, the ligation mixtures were directly transferred to competent DH5-α cells. Then, cells were plated, and colonies were picked and grown in LB medium with 100 μg mL$^{-1}$ carbenicillin for around 15 h. Afterward, plasmids were isolated using the GeneJET Plasmid Miniprep kit and verified by analytical digest with Nde1 and EcoR1 following gel electrophoresis. The DNA sequences of the inserts were verified by DNA sequencing (Microsynth Sequencing AG). We constructed pJET-prethrombin 2-SNAP and pJET-SUMO-Hirudin-SNAP in this manner. Finally, the fusion proteins were transferred into the expression vector pET25b(+) for protein expression (Supplementary Fig. 1).

### Protein expression, purification, and characterization

*Escherichia coli* (*E. coli*) BL21(DE3) was chosen as the expression strain in this work. For protein production, Terrific Broth medium containing phosphate buffer, glycerol, and supplemented ampicillin (100 μg mL$^{-1}$), was inoculated with an overnight culture to an initial optical density at 600 nm (OD600) of 0.1 or lower and incubated at 37 °C shaking at 200 rpm until OD600 reached 0.8-1.0. Then, protein production was induced with 1 mM IPTG overnight at 37 °C and 30 °C for prethrombin 2-SNAP and SUMO-Hirudin-SNAP, respectively. Cells were subsequently harvested by centrifugation (7000× g, 20 min, 4 °C), resuspended in a lysis buffer (50 mM sodium phosphate buffer, pH 8.0, 300 mM NaCl, 20 mM imidazole, 20 mM PMSF, 10 g mL$^{-1}$ DNaseI) and lysed by high-pressure homogenizer (Constant Systems Ltd.). Then, the prethrombin 2-SNAP and SUMO-Hirudin-SNAP were treated by different methods. As the expressed prethrombin 2-SNAP was present in inclusion bodies, the insoluble part was collected by centrifugation (15,000×g, 45 min, 4 °C). Then the pellet was washed using 1 M guanidine-HCl to remove contaminants. The washed inclusion bodies were resuspended and incubated in a buffer containing 7 M guanidine-HCl, 20 mM DTT and 1% Triton X-100. After this, the inclusion bodies were purified from the supernatant by Ni-sepharose chromatography. The supernatant was filtered using a 0.22 μm pore size membrane filter (Millipore Corp.), and loaded onto a Histrap fast flow column (General Electric), which was pre-equilibrated with a His-binding buffer (50 mM Tris Buffer, 500 mM NaCl, 5 mM Imidazole, 7 M Guanidine-HCl, 20 mM DTT and 1% Triton X-100, pH=8). Next, 5-8 column volumes of His-washing buffer (50 mM Tris Buffer, 500 mM NaCl, 25 mM Imidazole, 7 M Guanidine-HCl, 20 mM DTT, and 1% Triton X-100, pH=8) were added to remove impurities. Then 3 column volumes of His-elution buffer (50 mM Tris Buffer, 500 mM NaCl, 250 mM Imidazole, 7 M Guanidine-HCl, 20 mM DTT and 1% Triton X-100, pH = 8) were used to collect the target prethrombin 2-SNAP. After the solution with target protein was 10 times diluted, the refolding of the inclusion body was realized by gradient dialysis in the presence of refolding buffer (50 mM Tris buffer, 300 mM NaCl, 1% PEG (Mw = 8000 g mol$^{-1}$), 50 mM DTT, 1% Glycerol, 0.2 mM GSH, 0.1 mM GSSH). Then, the soluble prethrombin 2-SNAP was transferred into a suitable buffer and then treated with Factor Xa protease to obtain active Thr-S without further treatment before use.

For SUMO-Hirudin-SNAP purification, cell debris was removed by centrifugation (15,000×g, 30 min, 4 °C). Then, the protein was purified from the supernatant under native conditions by Ni-sepharose chromatography. Product-containing fractions were further treated with SUMO protease overnight at 4 °C. Then the product was purified again using a Ni-sepharose column.

For both proteins, the concentrations were determined by measuring the absorbance at 280 nm using a Microplate Reader Spectrophotometer (Molecular Devices SpectraMax M3). Protein purity was determined by sodium dodecyl sulfate-polyacrylamide gel electrophoresis (SDS-PAGE) stained with Coomassie staining solution and analyzed with ImageJ software. Photographs of the gels were taken with a Bio-Rad gel imager (E-box, Vilber). Matrix-assisted laser desorption induced-time of flight mass spectrometry (MALDI–TOF MS) was performed on a Bruker Daltonics ultrafleXtreme instrument. The samples were mixed 1:1 v/v with a matrix solution of 50 mg mL$^{-1}$ SDHB in TA50 solvent. The values determined by the mass spectrometry were in good agreement with the masses calculated from the amino acid sequences. Purified proteins were frozen in liquid nitrogen, lyophilized, and then stored at −80 °C until further use.

### Enzymatic assay

The fluorogenic thrombin substrate (Fluoro-s) was used to measure the enzyme activity. The substrate is cleaved by thrombin to produce a fluorescence signal at 450 nm. Before measurement, the purified prethrombin 2-SNAP was treated with Factor Xa protease overnight at 4 °C

to generate active Thr-S. At first, three groups (pure Fluoro-s, Fluoro-s + Thr-S, and Fluoro-s + Thr-S/Hir-S) were measured to verify the enzyme activity. Then, different molar ratios (1:0, 1:0.5, 1.1, 1:2, 1:5, 1:10) of Thr-S and Hir-S were added to the reaction solutions. The mixtures were incubated in a 96-well plate at room temperature for 30 min, and the fluorescence was measured using a Microplate Reader Spectrophotometer (Molecular Devices SpectraMax M5) to determine the enzyme activity and the best ratio for full thrombin inhibition.

Moreover, fibrinogen was also chosen as a substrate for an enzymatic essay. The increase in absorbance at 650 nm indicated the successful conversion of fibrinogen into fibrin fibers. Briefly, Thr-S (10 μl, 0.5 μM) was added to 180 μl of PBS buffer, and 10 μl of fibrinogen (10 mg mL$^{-1}$) was then added to the mixture. The absorbance intensity was monitored using the Microplate Reader Spectrophotometer every 1 min.

### ITC measurements
ITC measurements were performed using a MicroCal PEAQ-ITC (Malvern) at 25 °C. Protein samples were dissolved in PBS and degassed before titrations. The sample cell was filled with 270 μl of Thr-S (26.2 μM), and the syringe was filled with 70 μl of Hir-S (108 μM). Pure PBS was also injected into the reference cell to subtract the heat of the dilution. During each experiment, 0.4 μl of Hir-S was first injected into the sample cell, followed by 19 injections of 2 μl of Hir-S at intervals of 60 s. Once titration was completed, the raw titration peaks were integrated and analyzed by the instrument software. The calculated $K_d$ values were the average of triplicate measurements.

### Single-molecule force spectroscopy (SMFS) experiments
The SMFS experiments were performed on a commercial AFM instrument (NanoWizard-IV, JPK, JPK, Berlin, Germany), and cantilevers with a spring constant of 0.03 N m$^{-1}$ (MLCT, Bruker) were used. At first, the cantilevers and coverslips were cleaned and treated with chromic acid. Next, the cantilevers and coverslips were immersed in 1% (v/v, toluene) (3-aminopropyl)triethoxysilane (APTES) for amino silanization. NHS-PEG-NHS solution (1 mg mL$^{-1}$ in DMSO) was added to the cantilevers and coverslips and incubated for 1 h at room temperature to functionalize the surface with N-hydroxysuccinimide ester. Then, NH$_2$-BS was added to obtain BS-modified cantilevers and coverslips. Finally, Thr-S and Hir-S were attached to the BS on the coverslips and cantilevers for 1 h at room temperature, respectively. Both the coverslips and cantilevers were freshly prepared and used immediately. All the measurements were carried out in a PBS buffer at room temperature. Different stretching speeds were tested from 200 to 1600 nm s$^{-1}$. Typically, the cantilever was gently brought into contact with the functionalized surface (-800 pN), held on the surface for 2 s to ensure the binding of Thr-S and Hir-S, and then retracted at a constant speed. The force-extension curves were recorded at a sampling rate of 10 kHz using JPK data processing software and then further analyzed using a custom program in Igor 6.37 (Wavemetric Inc.).

### Hydrogel preparation
To prepare the mechanically enhanced hydrogels, the BS ligand was covalently conjugated to Thr-S and Hir-S by mixing them with a molar ratio of 1:1 on a shaker at 4 °C overnight. Then, the BS-linked Thr-S and Hir-S were mixed in water with a molar ratio of 1:2.5 to fully inhibit the thrombin activity. Afterward, PEGs (4-arm-PEG-SH$_{10k}$ wt%: PEGDA$_{10k}$, wt% = 1:2) and lithium phenyl-2,4,6-trimethylbenzoylphosphinate (LAP, 0.1%) and fibrinogen (1 *wt*.%) were dissolved in PBS buffers with fully inhibited thrombin-hirudin pairs (13/32.5 μM) as the precursor solution. The precursor solution was transferred into a custom-made mold with the desired thickness and shape and then exposed to UV irradiation (365 nm, 20 min). The as-prepared hydrogel samples were used for mechanical tests without any further treatment.

To prepare the mechanical loosened hydrogels, after BS modification through SNAP chemistry, sodium acrylate (40%), LAP (0.1%), thrombin-cleavable or non-cleavable peptide crosslinker (TCP linker, 0.5%), and the thrombin-hirudin pairs (13/32.5 μM) were chosen as the hydrogel precursor. After degassing, the precursor solution was poured into the mold, followed by UV irradiation for 20 min to fabricate the mechanically loosened hydrogels.

To prepare the bilayered hydrogels, the precursor solution 1 was first poured into the custom-made mold, covered with a piece of glass, and irradiated with UV light for 10 min to form a gel. Then, precursor solution 2 was added to the first hydrogel layer in the mold for further gelation. After being treated with UV for another 20 min, the bilayered hydrogel was obtained. We prepared three kinds of bilayered hydrogels: in the first one, one layer used the precursor solution of self-stiffening hydrogel or self-softening hydrogel, and the other layer used the same precursor but without the enzyme. These kinds of bilayered hydrogels were denoted as Stiffening/Normal or Softening/Normal bilayered hydrogels. We also used both self-stiffening and self-softening hydrogels as the two layers for the bilayered hydrogel preparation and denoted them as Softening/Stiffening bilayer hydrogels. All the prepared bilayer hydrogels were used for further tests without any further treatment.

### Mechanical characterization
All mechanical tests were performed using a Zwick Roell test Control II electronics system equipped with a 100 N load cell at a rate of 10 mm/min unless otherwise noted.

To verify the thrombin activation inside hydrogel networks, a fluorogenic substrate of thrombin was added to the hydrogel precursor. The fluorescence intensity of the as-prepared hydrogel samples was measured using confocal microscopy. Mechanical tests were carried out by stretching the hydrogels at a strain of 20%, 40%, 60%, 80%, 100%, 120%, 150%, 200%, 250%, and 300%, followed by maintaining the strain for 3 min to cleave the fluorogenic substrate. Then, the hydrogel samples were released to an unstretched state for further fluorescence measurements.

Cyclic tensile tests were also performed for the reversible thrombin activation. For every cycle, the hydrogel samples were stretched to a strain of 150% and maintained for 3 min. Then, the hydrogel samples were released to the unstretched state. The fluorescence intensity was monitored using confocal microscopy for every tensile cycle. Similar experimental parameters were also used for thrombin-enhanced/loosened cyclic tensile tests as well as for the shape-morphing bilayered hydrogels.

### Reporting summary
Further information on research design is available in the Nature Portfolio Reporting Summary linked to this article.

## Data availability
The data that support the findings of this study are provided in the source data file. Source data are provided in this paper. The DNA sequences of recombinant proteins (Thr-S, GenBank: OR965912; Hir-S, GenBank: OR965913) are available on GenBank. Any additional requests for information can be directed to and will be fulfilled by the corresponding authors. Source data are provided in this paper.

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

## Acknowledgements

L. Zheng acknowledges the funding from the National Nature Science Foundation of China (No. 22277018), the Zhejiang Provincial Natural Science Foundation for Distinguished Young Scholars (LR23B030001), and the Wenzhou Institute, University of Chinese Academy of Sciences (No. WIUCASQD2020015, WIUCASQD 2022006). A. Herrmann wishes to acknowledge financial support by the Werner Siemens-Foundation through the project TriggerINK, the DFG through SFB 985 'Functional Microgels and Systems' and the project 'Dynamic DNA Hydrogels'.

## Author contributions

L.Z. and A.H. supervised the research. L.Z., K.Z., and Y.Z. designed the experiments. K.Z., Y.Z., and C.H. expressed and purified the proteins. J.Z., X.W., and Y.C. performed AFM studies and analyzed data. K.Z. and X.G. synthesized and characterized all compounds. A.M. contributed to the microscopy measurement. K.Z. prepared all the hydrogels and performed the relevant studies. L.Z., A.H., K.Z., and Q.L. wrote the paper with input from all authors.

## Competing interests

The authors declare no competing interests.
