## [Peer Review File · Nature Communications]

Shape Morphing of Hydrogels by Harnessing Enzyme Enabled MechanoresponseREVIEWER COMMENTS

Reviewer #1 (Remarks to the Author):

First of all, the reviewer apologizes for late submission of the report due to an unexpected personal situation.

The authors report mechano-responsive hydrogels by utilizing enzymes. The gel contains complex of thrombin and hirudin, which are linked with the non-covalent interactions. While thrombin is inactive in the state of complex, when the gel is stretched, the complex is dissociated, and the embedded thrombin becomes active as an enzyme. The authors utilized this force-induced switching of the enzymatic activity of thrombin for creation of self-stiffening, self-softening, and self-morphing materials. The experiments have been designed very carefully with enough supporting information. The obtained active materials are very unique and novel with high scientific value. The reviewer suggests acceptance of this manuscript after some minor revisions.

1.

Sometimes the authors misuse the term “rate”. The reviewer has found,

Line 182: “extension rate” should be “extension ratio”

Line 242: “stretching rate” should be “strain”

Line 294: “extension rate” should be “strain” or “extension ratio”.

2.

For the morphing, the bending of the gel should occur after removal of the stretching force. Thus,

Line 295: “Upon stretching” should be “After removal of strain” or like that.

3.

The reason of the bending is unclear for the reviewer. The reviewer understands that stiffness of each layer changes upon deformation. However, if the length of each bilayer at its relaxed state is not changed, the bilayer material should keep the flat state despite change in the stiffness. Please explain the mechanism of bending more clearly.

4.

Since the enzymes are used, the authors are encouraged to add comments about stability and of the materials.

5.

Can the authors estimate dissociation ratio of the protein complex in the stretched gel? It would be possible to evaluate this by comparing the reaction rate of the stretched gel and that of the gel having free thrombin uncapped with hirudin.

6.

In the gel, the enzymes are anchored to the gel network and the reactants are also trapped in the gel

network, resulting their restricted diffusion in the gel. Thus, the reviewer suspects that the reaction coefficient of enzymatic reactions in this system would be remarkably smaller than those in solution due to the restricted diffusion. Can the authors add some comments about this?

7.

In figure 5e, stiffening and softening might be opposite.

Reviewer #2 (Remarks to the Author):

This manuscript describes the design of hydrogels that autonomously respond to repeated mechanical stimuli with either self-stiffening or self-softening. Unique building blocks of engineered thrombin and its inhibitor hirudin were designed, biosynthesized, and modified with reactive handles utilizing genetically encoded click chemistry for their integration into polymer networks, a PEG-based thiol-ene hydrogel or a poly(sodium acrylate) hydrogel. Fundamental calculations were performed to inform the design of these building blocks to enable the dissociation of the inhibitor from the enzyme in response to relevant levels of applied force, and experiments with relevant controls demonstrated how force could be repeatedly applied for the triggered cleavage of a model peptide. Subsequently, how repeated mechanical force could be applied to induce polymerization of fibrinogen for stiffening or cleavage of integrated peptide linkers for softening was demonstrated, and bilayer gels were then utilized to create 3D objects from 2D sheets exploiting these mechanisms. The work will be of interest to a broad audience and represents a significant advance in the field. Overall, the work is well done, and the manuscript is generally well written. There are few important points that need clarification as noted below.

1. Self-stiffening and self-softening?: As presented it is unclear if self-stiffening and self-softening can be done in the same material or sequentially. For example, Figure 4A shows self-stiffening, and Figure 4C shows self-softening. However, it is not clear to the reader from any of the figures that self-stiffening and self-softening can occur in the exact same material. If the approach is demonstrated for either self-stiffening or self-softening (but not both in the same material), then the wording needs to be adjusted throughout the manuscript to clearly delineate this point; for example, “and” changed to “or” in many places. If the approach supports both self-stiffening and self-softening in the same material in sequence, then the results and discussion need to be edited to demonstrate that point, including clarity / inclusion of data that clearly support it. The work is impactful either way, clarity is just needed.

2. Figure legends: the abbreviations used in the figure legends (e.g., Thr-S-HVI-S, Thr-S-BS-HVI-S-BS, etc) need to be defined in the figure captions. The reader can sort of figure out between the main text, figure, and caption that some of the conditions are controls; however, it is difficult in some cases to be sure what the abbreviation condition is vs. what is being described in the text. Defining the abbreviations in the caption (and potentially in the main text) will help the reader navigate the data more effectively.

3. Stiffening with fibrinogen (page 10): how is fibrinogen integrated within the hydrogel (e.g., during

hydrogel formation, diffused in; is it tether, entrapped, or can it freely diffuse)? Is the fibrinogen stable long-term within the hydrogel? Does the timing matter between when hydrogels are prepared vs. when mechanical force is applied for self-stiffening? How is the hydrogel with fibrinogen stored between preparation and the application of force (e.g., how stable is the system for autonomous function)? These points need to be clarified for understanding the utility of the approach.

4. Switch from PEG gel to Poly(sodium acrylate) gel (page 10): Why the work was switched from the PEG gel to the poly(sodium acrylate) gel needs to be clarified at the start of the paragraph for that portion of the work on page 10. Can the PEG gel not be softened? – the way the paragraph starts seems to imply that yet later the motivation is noted of wanting to demonstrate the effectiveness of the mechanism in a different gel system. Was the rationale that the PEG gels are for stiffening and the poly(sodium acrylate) gels are for softening building to the bilayer? Generally, within the context of the work shown, it needs to be clarified in what systems allow stiffening, what systems allow softening, and if those stiffening and softening can be done in the same system.

REVIEWER COMMENTS

Reviewer #1 (Remarks to the Author):

First of all, the reviewer apologizes for late submission of the report due to an unexpected personal situation.

The authors report mechano-responsive hydrogels by utilizing enzymes. The gel contains complex of thrombin and hirudin, which are linked with the non-covalent interactions. While thrombin is inactive in the state of complex, when the gel is stretched, the complex is dissociated, and the embedded thrombin becomes active as an enzyme. The authors utilized this force-induced switching of the enzymatic activity of thrombin for creation of self-stiffening, self-softening, and self-morphing materials. The experiments have been designed very carefully with enough supporting information. The obtained active materials are very unique and novel with high scientific value. The reviewer suggests acceptance of this manuscript after some minor revisions.

We thank the reviewer for appreciating the novelty and well-performed research presented in our work.

1. Sometimes the authors misuse the term “rate”. The reviewer has found,

Line 182: “extension rate” should be “extension ratio”

Line 242: “stretching rate” should be “strain”

Line 294: “extension rate” should be “strain” or “extension ratio”.

We thank the reviewer for pointing out our mistake of misusing the term “rate”. We have replaced them with the correct terms in our revised manuscript.

2. For the morphing, the bending of the gel should occur after removal of the stretching force. Thus, Line 295: “Upon stretching” should be “After removal of strain” or like that.

We thank the reviewer for pointing out the wrong expression. We have corrected it in our revised manuscript.

3. The reason of the bending is unclear for the reviewer. The reviewer understands that stiffness of each layer changes upon deformation. However, if the length of each bilayer at its relaxed state is not changed, the bilayer material should keep the flat state despite change in the stiffness. Please explain the mechanism of bending more clearly.

We thank the reviewer for the insightful comment. According to Timoshenko’s theory developed for bimetallic bilayer beams, the bending behavior of the bilayer hydrogel depends on several factors. As shown in Eq. (1), k is bilayer curvature, h is total thickness, $\Delta\alpha$ is the mismatch in the expansion coefficients, n is the stiffness ratio of both layers and m is the thickness ratio of both layers¹. We can see that the mismatch of two hydrogel layers is essential for the shape morphing behavior. In our bilayer hydrogel system, upon stretching the hydrogel, it triggers an enzymatic reaction that either densifies or loosens the network. After removal of mechanical force, both hydrogel layers tend to retract to an unstretched state. However, the mechanical properties of the two layers changed, which results in the discrepancy in retraction force between the two layers. Specifically the softer hydrogel layer exhibits less retraction compared to the stiffer hydrogel layer. This mismatch between the two layers is responsible for the observed deformation. In the picture of bending hydrogel in Fig 5e, we

can calculate that the length of softer part is around 17% longer than stiffer part.

$$k = \frac{\Delta\alpha}{h} \times \frac{6(1+m)^2}{3(1+m)^2 + (1+mn)(m^2 + \frac{1}{mn})}$$

(1)

4. Since the enzymes are used, the authors are encouraged to add comments about stability of the materials.

We thank the reviewer for the valid comment. To investigate the stability of the materials, specifically the stability of enzymes in hydrogel, we measured the reaction rates of thrombin using the fluorogenic substrate after storing the hydrogels for 0, 3, 7 and 14 days at 4 °C. As shown in Fig R1, the activity of thrombin decreased over time. After a period of 14 days, the activity of thrombin in the hydrogels had declined to below 50% compared to the initial activity of the freshly prepared hydrogels. These results indicate the occurrence of certain degrees of denaturation and/or degradation of embedded enzymes in hydrogels. Further research regarding the use of enzyme stabilizers or additives that protect the enzyme's structure and activity can help prolong its functional lifespan.

Fig. R1. Measurements of enzyme activity in solution and hydrogels that were stored at 4 °C for 0, 3, 7 and 14 days.

5. Can the authors estimate dissociation ratio of the protein complex in the stretched gel? It would be possible to evaluate this by comparing the reaction rate of the stretched gel and that of the gel having free thrombin uncapped with hirudin.

Following the reviewer's suggestion, we measured the reaction rates of stretched gel with free thrombin and hirudin-inhibited thrombin. By comparing the reaction rates shown in Fig. R2, the dissociation ratio of thrombin-hirudin complex at a stretching strain of 150% was estimated to be around 19%.

Fig. R2. The evaluation of dissociation ratio of thrombin-hirudin complex at a strain of 150%.

6. In the gel, the enzymes are anchored to the gel network and the reactants are also trapped in the gel network, resulting their restricted diffusion in the gel. Thus, the reviewer suspects that the reaction coefficient of enzymatic reactions in this system would be remarkably smaller than those in solution due to the restricted diffusion. Can the authors add some comments about this?

We appreciate the reviewer's valuable feedback. To investigate the impact of restricted diffusion within hydrogel networks on enzyme activity, we measured the reaction rate of thrombin in both PBS solution and hydrogels, As shown in Figure R1 (red and black lines), the thrombin activity within the hydrogels was found to be approximately 85% compared to the activity observed in the PBS solution. These results indicate that the restricted diffusion within the hydrogel network did not significantly affect the enzyme activity. One possible explanation for this observation is the nature of the fluorogenic substrate used in this study, which had a relatively small size (MW = 624 g/mol). However, when using larger enzyme substrates, we can anticipate a decrease in enzyme activity due to the limited diffusion within the hydrogel network. The larger size of the substrate may hinder its diffusion and slow down the rate at which it reaches the enzyme's active site, resulting in reduced enzyme activity.

7. In figure 5e, stiffening and softening might be opposite.

We thank the reviewer for pointing out the mistake. We have corrected it in the revised manuscript.

Reviewer #2 (Remarks to the Author):

This manuscript describes the design of hydrogels that autonomously respond to repeated mechanical stimuli with either self-stiffening or self-softening. Unique building blocks of engineered thrombin and its inhibitor hirudin were designed, biosynthesized, and modified with reactive handles utilizing genetically encoded click chemistry for their integration into polymer networks, a PEG-based thiol-ene hydrogel or a poly(sodium acrylate) hydrogel. Fundamental calculations were performed to inform the design of these building blocks to enable the dissociation of the inhibitor from the enzyme in response to relevant levels of applied force, and experiments with relevant controls demonstrated how force could be repeatedly applied for the triggered cleavage of a model peptide. Subsequently, how repeated mechanical force could be applied to induce polymerization of fibrinogen for stiffening or cleavage of integrated peptide linkers for softening was demonstrated, and bilayer gels were then utilized to create 3D objects from 2D sheets exploiting these mechanisms. The work will be of interest to a broad audience and represents a significant advance in the field. Overall, the work is well done, and the manuscript is generally well written. There are few important points that need clarification as noted below.

We thank the reviewer for giving high evaluation to our work.

1. Self-stiffening and self-softening?: As presented it is unclear if self-stiffening and self-softening can be done in the same material or sequentially. For example, Figure 4A shows self-stiffening, and Figure 4C shows self-softening. However, it is not clear to the reader from any of the figures that self-stiffening and self-softening can occur in the exact same material. If the approach is demonstrated for either self-stiffening or self-softening (but not both in the same material), then the wording needs to be adjusted throughout the manuscript to clearly delineate this point; for example, “and” changed to “or” in many places. If the approach supports both self-stiffening and self-softening in the same material in sequence, then the results and discussion need to be edited to demonstrate that point, including clarity / inclusion of data that clearly support it. The work is impactful either way, clarity is just needed.

We thank the reviewer for the valid argument. In the current work, self-stiffening and self-softening can be realized in different hydrogels, as we mentioned in the abstract: “Under cyclic tensile-loading, hydrogels exhibit self-stiffening or self-softening properties when substrates are present that can self-assemble to form new networks after being activated by thrombin or when cleavable peptide crosslinkers are constitutional components of the original network, respectively.” To address the reviewer’s concern, we changed “and” to “or” in many places in our revised manuscript.

2. Figure legends: the abbreviations used in the figure legends (e.g., Thr-S/HVI-S, Thr-S-BS-HVI-S-BS, etc) need to be defined in the figure captions. The reader can sort of figure out between the main text, figure, and caption that some of the conditions are controls; however, it is difficult in some cases to be sure what the abbreviation condition is vs. what is being described in the text. Defining the abbreviations in the caption (and potentially in the main text) will help the reader

navigate the data more effectively.

We thank the reviewer for the helpful comments. As the reviewer suggested, we have redefined the abbreviations in the figure captions.

3. Stiffening with fibrinogen (page 10): how is fibrinogen integrated within the hydrogel (e.g., during hydrogel formation, diffused in; is it tether, entrapped, or can it freely diffuse)? Is the fibrinogen stable long-term within the hydrogel? Does the timing matter between when hydrogels are prepared vs. when mechanical force is applied for self-stiffening? How is the hydrogel with fibrinogen stored between preparation and the application of force (e.g., how stable is the system for autonomous function)? These points need to be clarified for understanding the utility of the approach.

We appreciate the reviewer's insightful comment. As presented in the "Methods" section, fibrinogen was mixed with the hydrogel precursor and the resulting mixture was treated with UV to form hydrogels. To address the reviewer's concern regarding the state of fibrinogen within the hydrogels, we conducted additional experiments to explore its behavior. We integrated dye-modified fibrinogen into the hydrogels and allowed them to swell in PBS solution. As depicted in Figure R3, there were almost no fluorescent signals observed in the solution even after 24 hours, indicating that the fibrinogen is entrapped within the hydrogel.

Regarding the storage of hydrogels, we stored them in a sealed mold at 4 °C in a refrigerator after preparation. To examine the effect of hydrogel storage time on fibrinogen stability, we measured the self-stiffening property of the hydrogels after storage for 0-14 days. The results showed that the mechanical properties of the as-prepared hydrogels improved by approximately 70%, but then sharply decreased after day 7. This suggests that the fibrinogen remains relatively stable within the hydrogels for up to 7 days.

These findings provide evidence that the fibrinogen is entrapped within the hydrogel network and remains relatively stable for a certain period of time.

Fig. R3. Examination of fibrinogen state in hydrogels. (a). Confocal images for fluoro-labeled fibrinogen in hydrogels, left: bright field, right: dark field; (b). Fluorescent spectrum of fibrinogen contained hydrogels and PBS solution incubated with hydrogel for 24 hours.

Fig. R4. The evaluation of stability of fibrinogen in hydrogel through mechano-triggered self-stiffening property after storing for 14 days.

4. Switch from PEG gel to Poly(sodium acrylate) gel (page 10): Why the work was switched from the PEG gel to the poly(sodium acrylate) gel needs to be clarified at the start of the paragraph for that portion of the work on page 10. Can the PEG gel not be softened? – the way the paragraph starts seems to imply that yet later the motivation is noted of wanting to demonstrate the effectiveness of the mechanism in a different gel system. Was the rationale that the PEG gels are for stiffening and the poly(sodium acrylate) gels are for softening building to the bilayer? Generally, within the context of the work shown, it needs to be clarified in what systems allow stiffening, what systems allow softening, and if those stiffening and softening can be done in the same system.

We appreciate the reviewer's insightful comment. To achieve the self-softening property, a peptide crosslinker with a thrombin cleavage site was introduced during hydrogel preparation. This design allowed for the activation of thrombin upon stretching, which could then cleave the crosslinker and loosen the hydrogel network. While PEG-based hydrogel systems are also suitable for self-softening research, we considered the cost implications of replacing PEGDA with the thrombin cleavable peptide crosslinker. To address this, we decided to design a new hydrogel system that meets our research requirements while considering cost-effectiveness.

In designing the new hydrogel system, we took the following considerations into account. Firstly, we aimed to introduce monomers that are versatile and inexpensive as the hydrogel backbone. These monomers would be crosslinked using our peptide linker through UV-triggered polymerization. This approach allows for cost-effective hydrogel synthesis while maintaining the desired self-softening properties. Secondly, it was crucial to ensure that the monomers used in the new hydrogel system would not interfere with thrombin activity or react with the enzyme and inhibitor. This consideration ensures that the hydrogel system remains compatible with the enzymatic processes and maintains the desired functionality. These considerations allow us to explore the self-softening properties while minimizing expenses and maintaining the integrity of the enzymatic processes involved.

We added several sentences in the revised manuscript to clarify this issue.

REVIEWERS' COMMENTS

Reviewer #1 (Remarks to the Author):

The authors carefully read the comments from the reviewers and answered. While most of the answers are convincing, little revisions have been made to the main text and SI.

The reviewer made comments as one of potential readers of this manuscript. Thus, the authors are strongly encouraged to reflect the answers to the reviewers to the main text or SI, such as Figures R1-R4 and related explanations.

Reviewer #2 (Remarks to the Author):

The authors have addressed all of this reviewer's concerns in the revised manuscript. The work will be of broad interest.

REVIEWER COMMENTS

Reviewer #1 (Remarks to the Author):

The authors carefully read the comments from the reviewers and answered. While most of the answers are convincing, little revisions have been made to the main text and SI. The reviewer made comments as one of potential readers of this manuscript. Thus, the authors are strongly encouraged to reflect the answers to the reviewers to the main text or SI, such as Figures R1-R4 and related explanations.

Following the reviewer's suggestion, we have reflected the answers to the reviewers of the 1st round reviewing to the main text and SI. The added parts were highlighted in red in the revised manuscript, and the figures were added in SI.